# TEASER: Token Enhanced Spatial Modeling for Expressions Reconstruction

**Yunfei Liu[1]\***, **Lei Zhu[1,2]\*†**, **Lijian Lin[1]**, **Ye Zhu[1]**, **Ailing Zhang[1,2]†**, **Yu Li[1,‡]**
[1]International Digital Economy Academy
[2]Peking University, Shenzhen Graduate School
`{liuyunfei,zhulei,linlijian,zhuye,zhangailing,liyu}@idea.edu.cn`

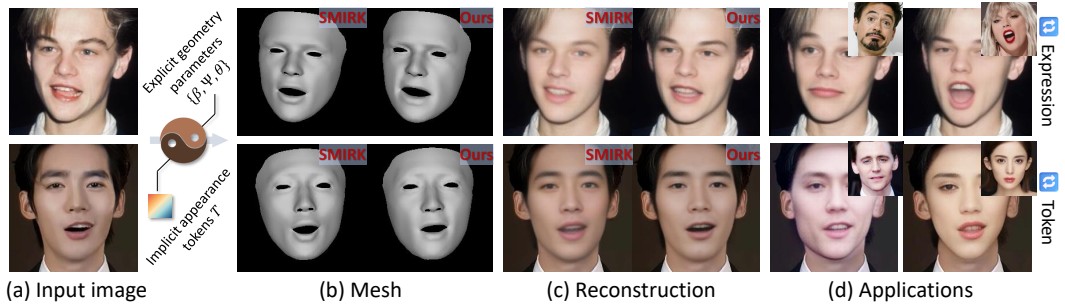

Figure 1: Given the input image (a), TEASER predicts hybrid parameters for 3D facial reconstruction. The explicit parameters can be used to reconstruct precise 3D facial expressions (b). The implicit parameters (*i.e.,* appearance token) guides high-fidelity face images generation (c). TEASER can be easily adapted to various applications, *e.g.,*, expression modification, as shown in the top row of (d), or changing facial appearance through token swapping, as shown in the bottom row of (d).

## Abstract

3D facial reconstruction from a single in-the-wild image is a crucial task in human-centered computer vision tasks. While existing methods can recover accurate facial shapes, there remains significant space for improvement in fine-grained expression capture. Current approaches struggle with irregular mouth shapes, exaggerated expressions, and asymmetrical facial movements. We present TEASER (Token EnhAnced Spatial modeling for Expressions Reconstruction), which addresses these challenges and enhances 3D facial geometry performance. TEASER tackles two main limitations of existing methods: insufficient photometric loss for self-reconstruction and inaccurate localization of subtle expressions. We introduce a multi-scale tokenizer to extract facial appearance information. Combined with a neural renderer, these tokens provide precise geometric guidance for expression reconstruction. Furthermore, TEASER incorporates a pose-dependent landmark loss to further improve geometric performance. Our approach not only significantly enhances expression reconstruction quality but also offers interpretable tokens suitable for various downstream applications, such as photorealistic facial video driving, expression transfer, and identity swapping. Quantitative and qualitative experimental results across multiple datasets demonstrate that TEASER achieves state-of-the-art performance in precise expression reconstruction. Code and demos are available at https://tinyurl.com/TEASER-project.

## 1 Introduction

3D facial reconstruction from a single image is a key component in several human-centered applications, including digital avatar creation, immersive telecommunication in AR/VR/XR, social

---

*Equal contribution.

†The work was done during Lei&Ailing's internship at the International Digital Economy Academy.

‡Corresponding author.

media, *etc.*Generally, these methods estimate the corresponding coefficients of the 3D Morphable Model (3DMM) Egger et al. (2020) for precise facial feature positioning and expression reconstruction. Due to the scarcity of large-scale paired data for single image-to-3D face mapping, many optimization-based methods Bas et al. (2017); Romdhani & Vetter (2005) or deep learning-based regression approaches Chang et al. (2018); Daněček et al. (2022); Deng et al. (2019b); Ploumpis et al. (2020); Tewari et al. (2017) commonly rely on self-supervised learning, adhering to the analysis-by-synthesis training paradigm.

Recent analysis-by-synthesis methods typically begin by using a network to estimate facial geometry, facial texture and environmental illumination from the input image. Then a differentiable renderer is employed to render a pseudo-image based on the extracted information. The loss between this pseudo-image and the input image is calculated as a supervision signal, allowing the network to be trained without requiring 3D data. Although these methods have achieved good shape reconstruction performance, they tend to fail in capturing facial expressions under certain scenarios, *e.g.,*, the input image contains extreme facial movements and expressions, including asymmetrical, exaggerated, or subtle motions perceptible to the human eye. The reasons are: 1) The differentiable renderer requires facial texture and environmental illumination. However, since the texture color on the face is often coupled with environmental illumination, estimating these two parameters simultaneously is challenging. 2) The pseudo-image generated by differentiable renderer is often overly smoothed, resulting in the loss of important facial details such as texture and expressions. And 3) the differentiable renderer struggle to model complex facial scenarios such as shadows, occlusions, and highly reflective facial surfaces, limiting the effectiveness of photometric loss. Some methods have attempted to constrain the 3DMM parameter search space by incorporating networks from other facial tasks, such as lip reading Filntisis et al. (2023), emotion estimation Daněček et al. (2022), face recognition Gecer et al. (2019), and GAN discriminators Otto et al. (2023), *etc.*However, these approaches provide indirect supervision for expression reconstruction and require a delicate balancing of weights among different loss components.

Recently, SMIRK Retsinas et al. (2024) introduced a UNet-based image generator to upgrade the traditional differentiable renderer. By circumventing the limitations of differentiable rendering, SMIRK improves the perceptual quality of reconstructed facial expressions. However, it heavily relies on randomly sampled pixels from the input images, resulting in low-quality outputs characterized by excessive smoothing and noticeable artifacts. Consequently, a considerable domain gap remains between the input and the synthesized images, as illustrated at $4^{th}$ column in Fig. 1.

Inspired by this approach, we propose Token EnhAnced Spatial modeling for Expressions Reconstruction (TEASER). Our method first extracts a hybrid representation combining explicit facial parameters (3DMM coefficients) and implicit appearance token. This token then guide a newly designed neural renderer to generate more faithful face images. Our novel neural renderer addresses issues in the neural renderer from SMIRK, thereby providing more accurate photometric loss and precise expression reconstruction. Furthermore, the extracted tokens not only enhance the stability of the neural renderer but also exhibit good interpretability, enabling TEASER to be applied to various downstream tasks including face editing, face animation, and identity swapping.

Specifically, TEASER introduces a multi-scale tokenizer that extracts complex facial representations from the input image, including self-occlusion shadows, glasses, lighting, and various skin tone, *etc.*, compressing them into a compact representation called *Token*. Subsequently, we design a Token-guided facial neural renderer that fuses Tokens at different scales to reconstruct high-fidelity input images based on facial mesh geometry. To train the tokenizer in an self-supervised manner, we propose a novel token cycle loss. To further improve the generalization of the method in subtle facial expression scenarios, we introduce a pose-dependent landmark loss and a region loss to improve facial expression details and face reconstruction performance in gaze/mouth areas, respectively.

In summary, the main contributions of this paper are as follows: 1) We propose TEASER, a novel approach that achieves more accurate facial expression reconstruction by predicting a hybrid representation of faces from a single image. 2) We design a multi-scale facial appearance tokenizer and introduce a token-guided neural renderer to generate high-fidelity facial images. The extracted token is interpretable and highly disentangled, enabling various downstream applications. 3) We develop a token cycle constraint for self-supervised training of the tokenizer. Additionally, we introduce pose-dependent landmark loss and region loss to further enhance the quality of expression reconstruction and facial image reconstruction. 4) TEASER achieves the state-of-the-art performance

including quantitative and qualitative results on multiple benchmark datasets. Rigorous experiments also demonstrate the efficiency of different components in TEASER. Furthermore, we showcase impressive results in various face editing and animation tasks.

## 2 RELATED WORK

### 2.1 3D FACE RECONSTRUCTION

Reconstructing 3D faces from 2D input images has received widespread attention over the past few decades Zollhöfer et al. (2018). Model-free approaches Deng et al. (2020); Dou et al. (2017); Feng et al. (2018a); Wu et al. (2020); Ruan et al. (2021) regress 3D vertices directly or optimize a Signed Distance Function Park et al. (2019) for image fitting. These techniques commonly require explicit 3D supervision during training, which limits their expressiveness due to inherent constraints in data creation and the differences between synthetic and real images Dou et al. (2017); Zeng et al. (2019).

With the development of statistical face models 3DMM, many methods for estimating coefficients of these models have emerged, employing a fixed linear shape space in an analysis-by-synthesis manner, such as BFM Paysan et al. (2009), FaceWarehouse Cao et al. (2013), FLAME Li et al. (2017b), 3DDFA-v3 Wang et al. (2024), *etc.*. Existing methods can be generally categorized into optimization-based Aldrian & Smith (2012); Bas et al. (2017) and learning-based approaches Chang et al. (2018); Kim et al. (2018); Zielonka et al. (2022b). Optimization-based methods require iterative optimization for each new image, which is time-consuming. With the rise of deep learning, learning-based methods have become mainstream, prompting many works to leverage various supervisory signals from different image domains, such as 2D keypoints Yang et al. (2020); Deng et al. (2019b); Shang et al. (2020) and 2D face contours Liu et al. (2017) for self-supervised training. However, for commonly used 2D key points, the sparsity and limited accuracy of the predicted points result in constrained supervision, particularly when facial expressions and head poses are complex. This often leads to misalignment between the 3D mesh and the input image. Photometric constraints are especially effective for image-domain data; however, they are vulnerable to alignment errors and rely on the quality of the rendered image.

To enable supervisory signals from the image domain to assist in reconstructing accurate 3D meshes, it is essential to obtain a precise representation of facial appearance or texture. Lin et al. (2020) enhances the initial 3DMM texture during the estimator training process, while Booth et al. (2018) utilizes a 3DMM for shape estimation, supplemented by a PCA appearance model learned from in-the-wild images. Gecer et al. (2019) expands on this concept by employing a GAN to model facial appearance more effectively. Additionally, Tewari et al. (2021); Tran et al. (2019) learns nonlinear models of shape and expression during the self-supervised training of a estimator.

Most of these studies generate renderings using linear statistical models and Lambertian reflectance Koppal (2020). In contrast, SMIRK introduces an innovative neural rendering module that tries addresses the domain gap between the input and the synthesized output. By reducing this discrepancy, SMIRK enhances the supervision signal within an analysis-by-synthesis framework. However, SMIRK heavily depends on randomly sampling some pixels from the source image. Although this approach ensures that facial structure information is not leaked, the supervision signal provided by the sampled pixels is not strong enough, resulting in significant differences between the reconstructed 2D image and the source image. The intermediate generated mesh also deteriorates accordingly, failing to align perfectly with the real image. We propose to use implicit appearance token that decouple from input image, representing semantic information such as facial texture and details, to provide stronger guidance, enabling the generalization of more accurate 2D images and corresponding meshes.

### 2.2 DISENTANGLED FACE REPRESENTATION LEARNING

The development of disentangled facial representation learning Chen et al. (2016); Higgins et al. (2016); Wei et al. (2021) has greatly benefited from advances in Generative Adversarial Networks (GANs) and self-supervised learning, particularly in the areas of image generation and facial editing. Early studies primarily focused on separating facial geometric structures from texture features. For example, Liu et al. (2015b); Wang et al. (2022); Yang et al. (2022) used an autoencoder model to disentangle identity and motion. While the 3D Morphable Model (3DMM) excels in modeling facial

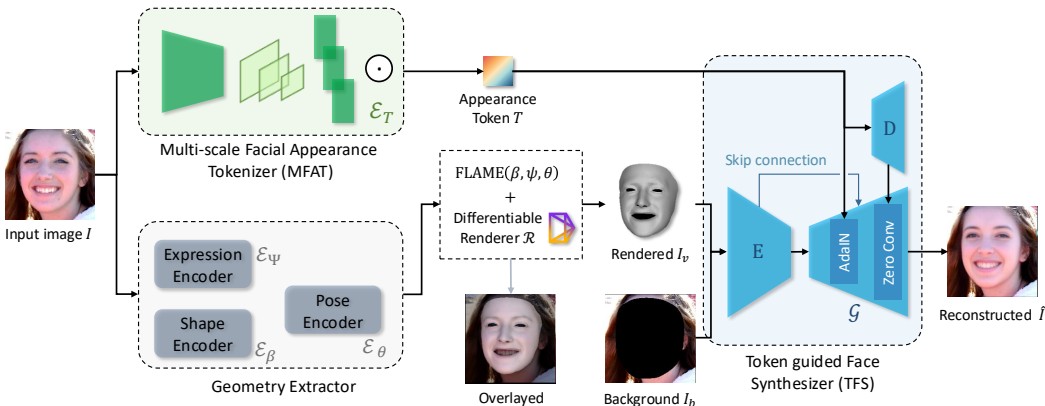

Figure 2: The framework of our pipeline.

geometry Ren et al., its texture modeling capabilities are limited, resulting in generated facial images that lack realistic texture details. This limitation arises mainly because the linear texture model of 3DMM is overly simplified. To address this, Tewari et al. (2017); Yin et al. (2022) combined 3DMM with GANs to generate high-quality textures, overcoming the shortcomings of traditional 3DMM. Deng et al. (2019b) further improved the detail and visual quality of the generated images by disentangling facial expressions, lighting, and textures. However, these methods face a dilemma: non-3DMM-based approaches struggle to model 3D-level information, while the facial textures included in 3DMM do not correspond to human perception. In this paper, we use 3DMM to obtain facial geometry information and facial tokenizer to derive high-level texture representations that align with human perception, achieving a balance between both approaches.

## 3 METHOD

This paper introduces TEASER, a novel approach that aims at extracting accurate hybrid representations from images, including explicit FLAME parameters for precise 3D facial expression geometry modeling and implicit tokens for capturing complex facial appearances. The overall network framework, as illustrated in Fig. 2, comprises three main components: 1) a Multi-scale Facial Appearance Tokenizer (MFAT) for capturing facial appearance token at different scales, 2) a Geometry Extractor that leverages existing techniques to extract parameters for FLAME modeling 3D geometric of facial expressions, and 3) a Token-guided Face Synthesizer (TFS) that generates facial images that are both aligned with the mesh image and rich in detail and fidelity, based on explicit facial mesh images and implicit tokens. We begin by introducing preliminaries and notation definitions in Sec. 3.1, followed by detailed descriptions of network modules, loss functions in subsequent subsections.

### 3.1 PRELIMINARY

Given an input image $I$, many existing methods employ FLAME to model the expressive 3D facial geometry. FLAME generates 3D vertices $V \in \mathbb{R}^{5023 \times 3}$ using shape coefficients $\beta \in \mathbb{R}^{300}$, expression coefficients $\psi \in \mathbb{R}^{50}$, jaw rotation $\theta_j \in \mathbb{R}^3$, head pose $\theta_h \in \mathbb{R}^3$, eye closure blendshapes $b \in \mathbb{R}^2$, and camera parameters $c \in \mathbb{R}^3$. In addition to geometry-related parameters, these methods typically predict appearance-related coefficients for facial modeling, including texture parameters $\gamma \in \mathbb{R}^{50}$ and scene lighting using Spherical Harmonics (SH) $l \in \mathbb{R}^{9 \times 3}$. $\gamma$ is used to generated the albedo $b \in \mathbb{R}^{5023 \times 3}$ for each vertex in $V$ through a FLAME-Tex model. To ensure the accuracy of extracted parameters, these method introduce self-supervised constraints between $I$ and reconstructed image $\hat{I}$, which is generated by a differentiable renderer based on Lambertian reflectance. Mathematically, the overall process is

$$\{\beta, \psi, \theta_j, \theta_h, b, c, \gamma, l\} = \mathcal{E}(I), \ \ V = \text{FLAME}(\beta, \psi, \theta_j, \theta_h, b), \quad \text{(1),(2)}$$

$$C = \text{FLAME-Tex}(\gamma), \ \ \hat{I} = \mathcal{R}(V, C, l, c). \quad \text{(3),(4)}$$

Unlike previous methods that predict facial texture PCA coefficients $\gamma$ and simplified environmental lighting $l$, TEASER introduces an implicit latent code, namely token $T$, to represent complex facial appearance. Simultaneously, TEASER incorporates a novel neural renderer $\mathcal{R}^*$ that takes both explicitly rendered mesh image and implicit token, generating high-fidelity facial images that bridge the domain gap between the input and synthesized output $\hat{I}$. Consequently, we re-write Eqn. 1, Eqn. 3 and Eqn.4 as follows:

$$\{\beta, \Psi, \theta, T\} = \mathcal{E}(I), \tag{5}$$

$$\hat{I} = \mathcal{R}^*(I_v, I_b, T), \text{ where } I_v = \mathcal{R}(V, C_0, l_0, c). \tag{6}$$

Here, for brevity, we combine jaw pose and expression parameters into a joint expression representation $\Psi = \{\psi, \theta_j, b\}$. We also combine camera motion and head pose to global transformation parameters, *i.e.,* $\theta = \{c, \theta_h\}$. $C_0$ and $l_0$ are predefined vertices color and environmental lighting, respectively. $I_b$ is the background image where the face area is masked out. Following previous work, we treat head pose $\theta_h$ as a rigid transformation and use a weak perspective camera model.

## 3.2 Hybrid Facial Representation

**Multi-scale Appearance Tokenizer.** We design a Multi-scale Appearance Tokenizer (MFAT, $\mathcal{E}_T$) to capture appearance features with rich facial appearance information. Given an input image $I$, we first employ a multi-layer CNN-based image encoder $E$ to extract image features. The extracted features at different layers contain varying levels of semantic information. Thus, we propose to use all these features to capture both high-dimensional semantics and fine-grained texture details for a more comprehensive facial representation. Specifically, we adopt an average pooling layer $\mathcal{P}$ followed by a fully connected layer $\mathcal{F}$ after each layer to project these features into a unified space. Finally, we concatenate these multi-scale features together as a comprehensive facial appearance token $T$. The computational process can be formulated as follows:

$$T = \mathcal{E}_T(I) = z_1 \odot \cdots \odot z_K, \tag{7}$$

where $\odot$ is channel-wise concatenation, $z_i = \mathcal{F}_i(\mathcal{P}(x_i)), i \in \{1, \cdots, K\}$. $x_i$ is the feature map from $i$-th stage in image encoder $E$. By doing this, our appearance token $T$ incorporates multi-level information from the input image, resulting in a more detailed and precise facial representation.

**Facial Geometry Extractor.** To extract the geometric information from the input image, we follow Retsinas et al. (2024) and use three identical encoders $\mathcal{E}_\beta, \mathcal{E}_\Psi, \mathcal{E}_\theta$ to predict the face shape $\beta$, expression $\Psi$, and transpose parameters $\theta$, respectively. Formally,

$$\beta = \mathcal{E}_\beta(I), \quad \Psi = \mathcal{E}_\Psi(I), \quad \theta = \mathcal{E}_\theta(I). \tag{8}$$

## 3.3 Token guided Face Synthesizer

We argue that an effective neural renderer should possess two key characteristics: a) The generated facial images must maintain strict alignment with the facial geometry (*i.e.*, the rendered mesh image). This alignment provides accurate spatial guidance for expression geometry reconstruction when computing self-supervised losses. b) The generated faces must be of high fidelity, minimizing the gap with the input image to reduce errors associated with photometric loss. To achieve these objectives, we introduce a Token-guided Face Synthesizer (TFS). Our innovation lies in implementing a facial neural rendering network controlled by hybrid information, combining explicit face geometry and implicit appearance tokens. Specifically, our input contains three parts: 1) face geometric information $I_v$, which is generated from differentiable rasterization step on the reconstructed FLAME vertices $V$, as provided at Eqn. 6. 2) Facial appearance token $T$, which is extracted from $\mathcal{E}_T$. $T$ implicitly contains skin colors, face texture, complex environment illumination, *etc.*And 3) Non-face background $I_b$. We mask out the face part to obtain the background region. Based on these inputs, the generation process of our TFS can be represented as:

$$\hat{I} = \text{TFS}(I_v \odot I_b, T). \tag{9}$$

$\hat{I}$ represents the generated image, which follows the geometric information of $I_v$, incorporates the appearance of $T$, and uses $I_b$ as the background. TFS is built on a U-Net structure for multi-scale

feature extraction, which firstly takes face geometric $I_v$ and the non-face background $I_b$ as input. As for the appearance token $T$, we adopt two techniques to incorporate $T$ into TPS to guide the face generation process. Initially, we adopt adaptive instance normalization (AdaIN Huang & Belongie (2017)) to modulate the generation with $T$, which provides global style guidance to TPS. To better inject high-frequency details, inspired by ControlNet Zhang et al. (2023), we create a new token decoder $D$ by referring to the decoder, where each block is followed by a zero-convolution layer. Each feature map extracted by each block of $D$ is added to the features from the UNet decoder. Combining AdaIN and token decoder $D$, TPS effectively captures both global style information and detailed high-frequency features, leading to more refined face generation results with rich facial details. For the detailed network architecture please refer to the supplementary materials.

## 3.4 Loss functions

To reconstruct 3D facial mesh from image $I$, we design a learning framework to minimize the total loss $\mathcal{L}$ as follows:

$$\mathcal{L} = \lambda_{ec}\mathcal{L}_{ec} + \lambda_{lmk}\mathcal{L}_{lmk} + \lambda_{tc}\mathcal{L}_{tc} + \lambda_{pdl}\mathcal{L}_{pdl} + \lambda_{rg}\mathcal{L}_{rg} + \lambda_{ic}\mathcal{L}_{ic}, \tag{10}$$

where $\mathcal{L}_{ec}$ is the augmented expression consistency loss, aiming to reduce over-compensation errors and promoting diverse expressions, we borrow it from Retsinas et al. (2024). $\mathcal{L}_{lmk}$ is the landmark loss, which is inherit from Zielonka et al. (2022a).

**Token Consistency Loss.** Due to the lack of ground-truth tokens, we adopt a token consistency loss $\mathcal{L}_{tc}$ to conduct self-supervised training in a cycle training manner. Given an input image $I$, we first extract its appearance token $T$ and geometric parameters $\beta, \Psi, \theta$. To enhance the stability of tokens across diverse expressions, we propose to augment the expression parameter $\Psi$ to $\Psi_{aug}$ by randomly changing certain values. The augmented $\Psi_{aug}$ and the original facial features are fed into our TFS to generate face images $\hat{I}_{aug}$ with different expressions to $I$. Finally, the $\mathcal{L}_{tc}$ computes the mean-squared error between the original Token $T$ with the token predicated on $\hat{I}_{aug}$:

$$\mathcal{L}_{tc} = \sum_{i=1}^{N} \left\| \mathcal{E}_T(\text{TFS}(\mathcal{R}(V_i', C_0, l_0, c) \odot I_b, T)) - T \right\|_2, \tag{11}$$

where $V_i' = \text{FLAME}(\beta, \Psi_{aug}, \theta)$, $N$ denotes the number of times we randomly augment $\Psi_{aug}$. This loss guarantees the predicted token to be consistent under different expressions.

**Pose-dependent Landmark Loss.** We observed that commonly used facial landmark detection methods Feng et al. (2021); Zielonka et al. (2022a) in previous approaches often yield inaccurate results, particularly in distinguishing the boundary between the lower lip and lower teeth, or when dealing with complex expressions. To address this, we employ a state-of-the-art 2D facial landmark detector that accurately captures facial features under challenging conditions, including complex expressions, large head poses, and asymmetrical mouth shapes.

To tackle the issue of non-correspondence between 2D and 3D cheek contour landmarks due to pose variations, we deviate from conventional dynamic landmark marching Zhu et al. (2015). Instead, we introduce a novel pose-dependent landmark loss that adapts to the specific head orientation $\theta$, ensuring more robust and accurate facial reconstruction across diverse poses.

$$\mathcal{L}_{pdl} = \|M_L(\theta_y)L - M_L(\theta_y)L_v\|_2, \tag{12}$$

where $L \in \mathbb{R}^{203 \times 2}$ and $L_v \in \mathbb{R}^{203 \times 3}$ are 2D landmarks detected from $I$ and 3D landmark selected from face mesh $V$, respectively. $M_L$ is a pose-dependent mask, which is defined as

$$M_L(\theta_y) = \mathbb{1}_{\theta_y < -\epsilon} M_L^{left} + \mathbb{1}_{\theta_y > \epsilon} M_L^{right} + \mathbb{1}_{-\epsilon \leq \theta_y \leq \epsilon} M_L^{front}, \tag{13}$$

where $\mathbb{1}$ is indicator function, $\theta_y$ denotes the yaw angle of head pose, $\epsilon$ is a tunable parameter that determines the range of frontal face. We empirically set $\epsilon = 0.05$ in our experiments. $M_L^{left}, M_L^{right}, M_L^{front}$ are masks for visible landmarks under different head poses.

**Region Loss.** Previous neural facial renderer Retsinas et al. (2024) tends to generate faces with misaligned gaze information, lack of wrinkle details, and color shifts in the lips. Although these complex representations can be naturally addressed through our extracted implicit appearance tokens,

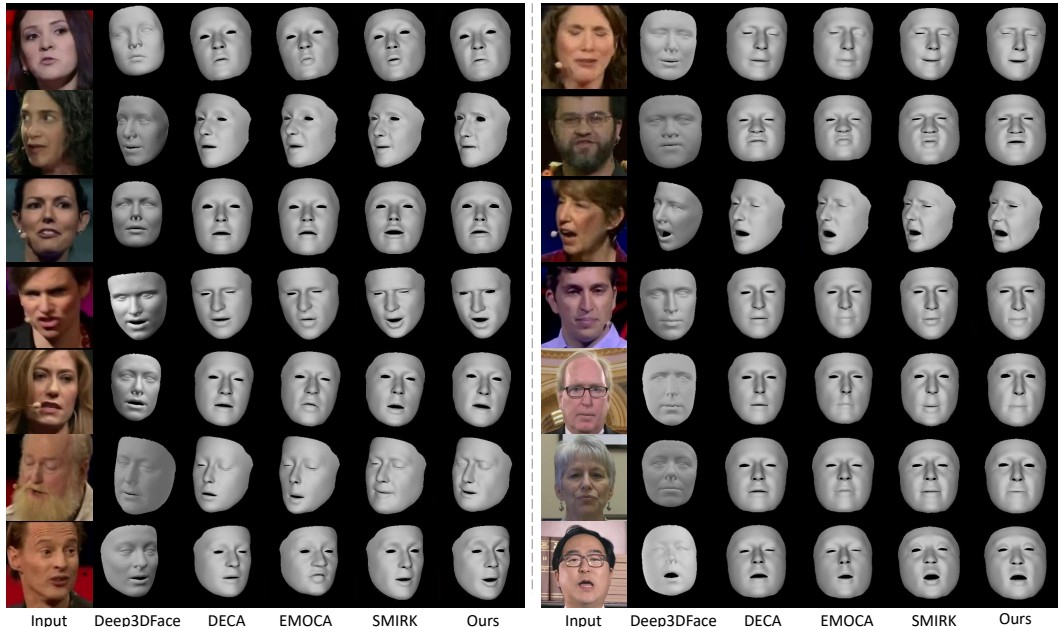

Figure 3: Visual comparison of 3D face reconstruction with SOTA methods.

the generated faces still exhibit minor discrepancies compared to the input images. We attribute this issue to the small size of these perceptually sensitive regions, making it challenging for the neural renderer to learn. To address this, we employ a masking strategy to calculate the local loss in the mouth and eye areas. Specifically, we parse facial landmarks to obtain masks for the eye region ($M_e$) and mouth region ($M_m$).The formula for this landmark-guided region loss is as follows:

$$\mathcal{L}_{rg} = \left\| (M_m + M_e) \cdot I - (M_m + M_e) \cdot \hat{I} \right\|_2. \tag{14}$$

**Photometric Loss and Perceptual Loss.** Similar to previous approaches Deng et al. (2019b); Feng et al. (2021), we apply common image-level supervision, including photometric loss $\mathcal{L}_{pho}$ and VGG perceptual loss $\mathcal{L}_{pho}$, to ensure consistency between the generated image $\hat{I}$ and the input image $I$:

$$\mathcal{L}_{ic} = \lambda_{pho}\mathcal{L}_{pho} + \lambda_{per}\mathcal{L}_{per}. \tag{15}$$

## 4 EXPERIMENT

### 4.1 EXPERIMENTAL SETUP

**Training Datasets.** We use the following datasets for training: FFHQ Karras et al. (2019), CelebA Liu et al. (2015a), and LRS3 Afouras et al. (2018). Since LRS3 is video dataset, we randomly sample images from each video frames from during training. We crop the frames based on mediapipe landmark  and resize them to the resolution of $224 \times 224$. We follow Retsinas et al. (2024) and separate different videos for training and testing.

**Implementation Details.** Our model is implemented in PyTorch Imambi et al. (2021). All models are trained on one NVIDIA RTX 3090 GPU and the batchsize is 16. We use MobileNet-V3 Howard et al. (2019) as the image encoder in MFAT for the balance between performance and efficiency. We use a learning rate of 0.001 to train our model with the Adam optimizer. We set the number of scales in MFAT to 4 and set the dimension of all tokens to 256. In our loss function, we set $\lambda_{ec} = 1.0, \lambda_{lmk} = 100, \lambda_{tc} = 5.0, \lambda_{rg} = 10.0, \lambda_{ic} = 10.0, \lambda_{pdl} = 500.0, \lambda_{pho} = 1.0, \lambda_{per} = 1.0$. We utilize a pre-trained geometry encoder from MICA Zielonka et al. (2022a). During the training process, $\mathcal{E}_\beta$ and $\mathcal{E}_\theta$ remain frozen to maintain stability and leverage the robustness of the pre-trained components. We alternately train the encoders (including the geometry encoder and MFAT) and TFS

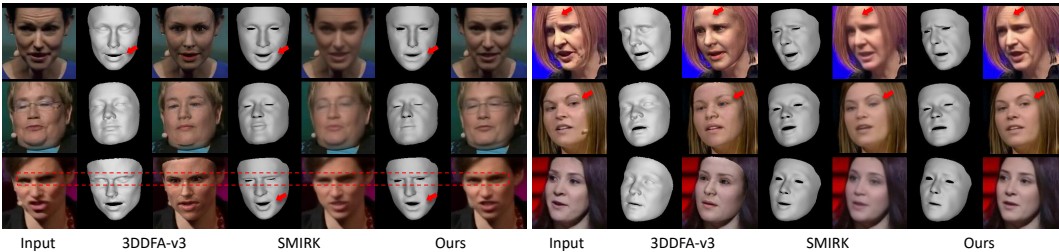

Input        3DDFA-v3        SMIRK        Ours        Input        3DDFA-v3        SMIRK        Ours

Figure 4: Visual comparison of estimated expression and its corresponding reconstructed images.

during the training process. This design helps reduce the effect of the neural renderer compensating for the encoders. For more training details please refer to the supplemental materials.

**SOTA Methods.** We compare with the following publicly available methods, including DECA Feng et al. (2021), EMOCA Daněček et al. (2022), and SMIRK Retsinas et al. (2024), which use the FLAME model Li et al. (2017a). We also involves the comparisons with Deep3DFace Deng et al. (2019b) and 3DDFA-v3 Wang et al. (2024), which use BFM model Paysan et al. (2009).

**Evaluation Metrics.** Recent methods Daněček et al. (2022); Retsinas et al. (2024) have reported that using 3D facial geometry to measure the accuracy of facial expression reconstruction is ill-posed. Therefore, we evaluate the quality of final reconstructed 2D images and videos. The insight is that if the quality of these reconstructed 2D images is high, then in our alternative training mode, the intermediate generated mesh will also align more closely with the input image. We use the Peak Signal-to-Noise Ratio (**PSNR**), Fréchet Inception Distance (**FID**) Heusel et al. (2017), cosine similarity of identity (**CSIM**), Average Expression Distance (**AED**), and Average Pose Distance (**APD**) to evaluate image-level reconstruction performance. We use warp-error Geyer et al. (2023) and calcute flicker level between adjacent frames to evaluate video-level reconstruction. For more details please refer to the supplementary materials.

Table 1: Quantitative comparisons of the reconstructed image on LRS3 and HDTF test dataset.

|  | Methods | Images | | | | | | Videos | |
|---|---|---|---|---|---|---|---|---|---|
|  |  | LPIPS ↓ | FID ↓ | CSIM ↑ | PSNR ↑ | AED ↓ | APD ↑ | warp-error↓ | flicker ↓ |
| LRS3 | SMIRK | 0.109 | 25.39 | 0.729 | 29.14 | 0.147 | 0.056 | 1.245 | 1.588 |
|  | 3DDFA-V3 | 0.181 | 56.18 | 0.604 | 25.78 | 0.141 | 0.054 | 0.903 | 1.372 |
|  | Ours | **0.077** | **19.41** | **0.804** | **30.67** | **0.114** | **0.044** | **0.801** | **1.167** |
| HDTF | SMIRK | 0.114 | 35.27 | 0.732 | 27.39 | 0.138 | 0.125 | 1.958 | 1.484 |
|  | 3DDFA-V3 | 0.399 | 104.90 | 0.597 | 12.87 | 0.144 | 0.068 | 1.736 | 0.870 |
|  | Ours | **0.081** | **30.19** | **0.826** | **28.35** | **0.036** | **0.029** | **1.422** | **0.662** |

## 4.2 Qualitative and Quantitative Results

### 4.2.1 Quantitative Comparison

We evaluate the image quality of our reconstructed images in two test datasets to validate the effectiveness of our method and the result is shown in Tab.1. Compared with 3DDFA-v3 and SMIRK across two benchmark datasets, our method demonstrates a significant advantage over the two methods. Specifically, higher LPIPS and lower FID values indicate that the perceptual difference between the 2D images reconstructed by our method and the input images is smaller. TEASER gets the highest CSIM value, suggesting that the extracted tokens contain rich facial appearance information. The better AED and APD results indicate that the expressions and head poses in our reconstructed images are more accurate. In the evaluation of videos, the reduction in warp-error and flicker values demonstrates that our generated videos exhibit better stability. This improvement is attributed to the high consistency of our tokens compared to the randomly sampled pixels in SMIRK, allowing for smoother transitions between video frames, which plays a crucial role in video generation.

### 4.2.2 Qualitative Comparison

**3D Mesh Accuracy.** Fig. 3 presents a visual comparison of the 3D meshes from different methods. Our approach, along with SMIRK, outperforms Deep3DFace, DECA, and EMOCA in capturing more vivid facial expressions. However, our method excels over SMIRK by producing more precise

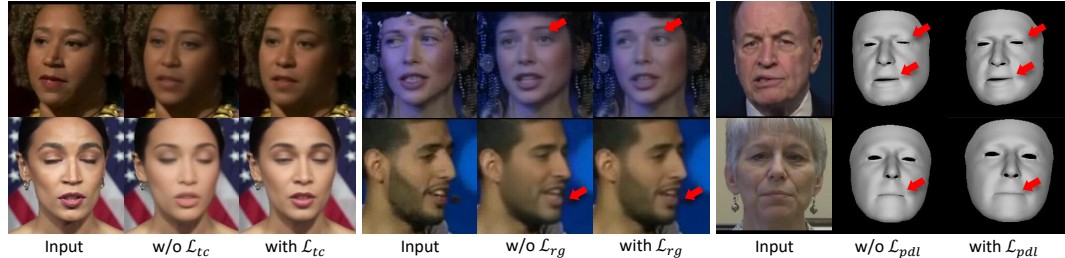

Input    w/o $\mathcal{L}_{tc}$    with $\mathcal{L}_{tc}$    Input    w/o $\mathcal{L}_{rg}$    with $\mathcal{L}_{rg}$    Input    w/o $\mathcal{L}_{pdl}$    with $\mathcal{L}_{pdl}$

Figure 5: Visual results of ablation study. Left: impact of token consistency loss. Middle: impact of region loss. Right: impact of proposed landmark loss.

Table 2: Ablation study on the LRS3 test dataset.

| Version | Description | LPIPS ↓ | FID ↓ | CSIM ↑ | PSNR ↑ | AED ↓ | APD ↓ |
|---|---|---|---|---|---|---|---|
| V1 | Baseline | 0.109 | 25.39 | 0.729 | 29.14 | 0.147 | 0.056 |
| | V1 (+ token $T'$) | 0.095 | 25.05 | 0.723 | 28.39 | 0.144 | 0.055 |
| V2 | V1 (+ token $T$) | 0.089 | 20.46 | 0.740 | 28.86 | 0.137 | 0.054 |
| V3 | V2 (+ $\mathcal{L}_{tc}$.) | 0.087 | 19.96 | 0.755 | 29.04 | 0.133 | 0.054 |
| V4 | V3 (+ $\mathcal{L}_{rg}$) | 0.086 | 20.30 | 0.762 | 29.17 | 0.123 | 0.046 |
| V5 | V4 (+ Zero Conv.) | 0.078 | 19.11 | 0.798 | 30.51 | 0.118 | 0.045 |
| Final | V5 (+ $\mathcal{L}_{pdl}$) | **0.077** | **19.41** | **0.804** | **30.67** | **0.114** | **0.044** |

and nuanced expressions. Notably, our method excels in modeling subtleties such as the mouth corner movements, intricate mouth shapes, and degree of eye openness.

**Reconstruction Facial Image Quality.** Fig.4 visually compares the reconstructed image with 3DDFA-v3 and SMIRK. It can be seen that the quality of our reconstructed images far surpasses these two SOTA methods. There are significant improvements in image clarity, appearance details (including eye gaze, wrinkles, sharpness of teeth), and the alignment of lip shape and color. Additionally, we show that our 3D mesh and 2D reconstructed images are almost perfectly aligned.

## 4.3 ABLATION STUDY

**Ablation on the Significance of Appearance Token.** In order to fully demonstrate the effectiveness of the facial tokens, we first conducted an ablation study on them. We take SMIRK as the baseline model. As shown in Fig.5, the tokens enhance the reconstructed 2D image in three aspects. First, the overall clarity of the image is improved. Second, the shape of the mouth on the face gradually approaches the original image. Additionally, compared to the baseline, there is a significant increase in gaze accuracy and teeth clarity, resulting in more accurate meshes (specifically reflected in the position of the eye contours and the shape of the mouth). We also remove multi-scale strategy in MFAT, and use only the final output of the token encoder as the facial appearance token, denoted as $T'$. It is worth noting that we removed the mask from the baseline that could leak facial color information, resulting in a slight decrease in both PSNR and CSIM metrics. As seen in Tab.2, the inclusion of the token significantly and consistently improves other evaluation metrics.

**Effect of the Token Consistency Loss and Region Loss.** As shown in the second row of Fig.5 and Tab.2. Compared to the baseline, our token have learned more information representing facial details without adding any constraints. However, there are still some shortcomings, as seen in the third column in Fig. 5, such as incorrect gaze and over smooth teeth. After adding region loss, the gaze and the density of the beard in the reconstructed images show significant improvement. This is because under the constraint of region loss, our tokens pay more attention to the mouth and eye areas, and under the constraint of cycle loss, the information extracted by the tokens is closer to the input image, thus improving fidelity.

**Effect of Pose-dependent Landmark Loss.** In order to further enhance TEASER's performance, including modeling complex mouth shape and improving the accuracy of mouth closure issue, we introduce pose-dependent landmark loss. In the third row of Fig.5, $\mathcal{L}_{pdl}$ compensates for a minor flaw that was present before its addition, such as more accurate lip closure.

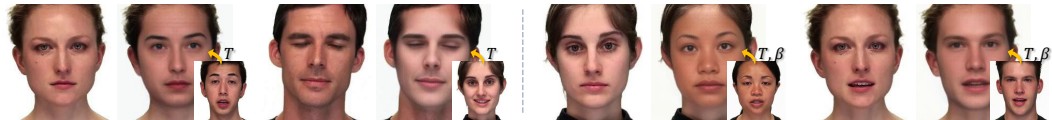

Figure 6: Application of our token in face editing. Left: we only replace tokens $T$ in the first row. Right: we replace both tokens $T$ and the FLAME shape parameters $\beta$.

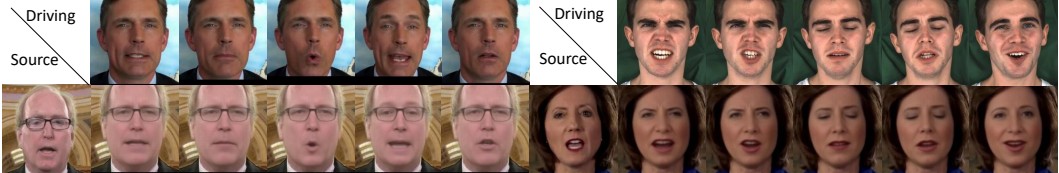

Figure 7: Application of our token in expression transfer.

## 4.4 TOKEN ANALYSIS

**Token Interpretability via t-SNE Clustering Visualization.**

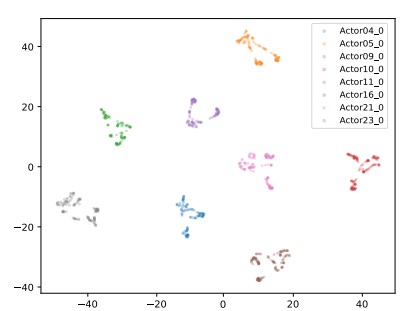

To validate our motivation, we investigate the multi-scale tokens that we extracted. We extract multi-scale tokens from each frame of videos of several different individuals, and perform clustering on these tokens across various scales. The clustering results are presented in Fig.8. We mark the tokens from the same person with the same color, and it can be seen that tokens from the same individual coalesce into distinct groups, with clear boundaries between these groups, and this is the case for all four scales of tokens. Given that the expressions and head poses vary in every frame within the same video, the ability to cluster them supports our hypothesis that *the tokens encapsulate identity-aware facial texture information*. The slight variations within the same category of tokens are attributed to the changes in texture that occur when expressions change.

Figure 8: Different subjects' tokens visualization via t-SNE.

**Token Transfer for Identity Swapping and Face Animation.** Since our tokens can represent facial textures, we apply them to the task of face swapping, with the results shown in Fig. 6. In the top row of Fig. 6, we only transferred the target's token to the source, and it can be observed that the swapped face retains the expressions, poses, and shapes consistent with the source, while the textures, such as skin and eyebrow color, are taken from the target. To more closely align with the definition of face swapping, we transferred the target's tokens and the FLAME shape parameters to the source. As shown in the bottom row of Fig. 6, the resulting swapped identity is almost identical to the target, while the expressions and other non-identity information remain consistent with the source.

## 5 CONCLUSION

In this paper, we propose a hybrid representation approach to enhance the accuracy of facial expression reconstruction. Our method designs a multi-scale facial appearance encoder to extract implicit facial appearance tokens, while employing a geometry encoder to capture explicit facial geometry information. In addition, we introduce a token-guided facial neural synthesizer, which generates accurate and stable facial images based on the spatial information from facial meshes. This provides finer and more precise supervision signals for facial expression reconstruction, enabling the capture of detailed facial expressions from a single image. Our results demonstrate the interpretability of the implicit tokens and highlight several potential applications, holding promising implications for visual effects and other human-centered tasks. Overall, we contribute a novel facial expression extraction method and a hybrid signal-driven facial generation network, paving the way for future tasks in complex 3D facial capture and tracking.

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

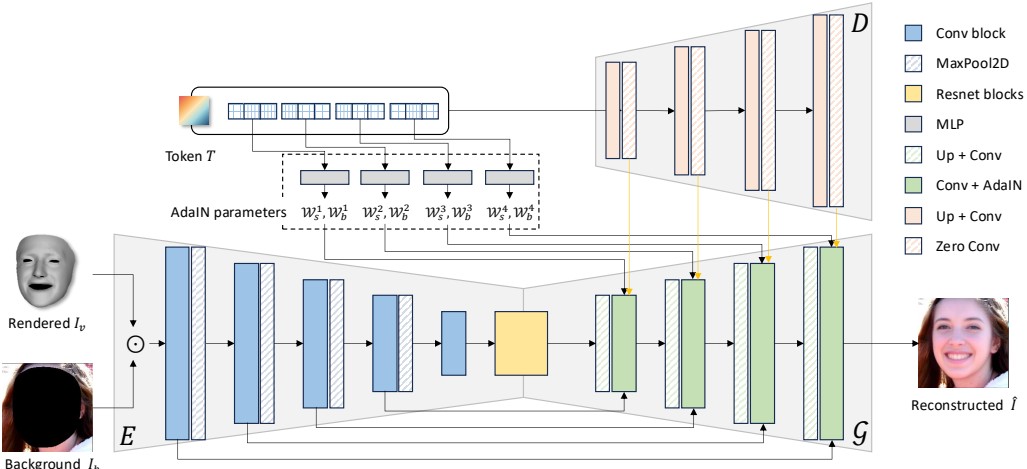

Figure A-1: Detailed architecture of Token-guided Face Synthesizer.

## A OUTLINE

This supplementary document provides more analyses and results that were not included in the main paper due to space limitations. The contents are organized as follows.

- Section B: Detailed architecture of our Token-guided Face Synthesizer.

- Section C: More implementation details about training, evaluation and loss design.

- Section D: More results on different 3D benchmarks, ablation studies, network parameters analysis, and inferecne speed details.

- Section E: Detailed explanation and analysis of our multi-scale appearance token and more cluster results of appearance token on different person videos.

- Section F: More discussions about our limitations, future work and ethical considerations.

## B ARCHITECTURE DETAILS

**Token-guided Face Synthesizer.** As shown in Fig A-1, the proposed TFS consists of a UNet-based face generator (*i.e.,*, an encoder $E$, a bottleneck block, and a generator $\mathcal{G}$), 4 MLP-layers, and a 4-layer token decoder. There are 4 blocks in $E$ and $\mathcal{G}$. Each block contains an up/down-sampling layer and two convolutional layers. Given a rendered mesh $I_v$ and a background $I_b$, the encoder $E$ first extracts multi-level features. These features are fed into the generator $\mathcal{G}$ by skip-connection. To better inject the appearance token $T$ to guide the generation process, we use two ways. Specifically, we first use AdaIN to modulate each block in $\mathcal{G}$, which provides a global style guidance for TPS. According to Sec 3.2 in the main paper, $T$ consists of $K$ ($K = 4$) sub-tokens. In TFS, we adopt $K$ MLP-layers to map these $K$ sub-tokens into AdaIN normalization parameters. Each of these parameters is fed into different layers in $\mathcal{G}$ to modulate features from $\mathcal{G}$, respectively. Meanwhile, we design a multi-layer token decoder $D$, where each block is followed by a zero-convolution layer. The token decoder $D$ takes the whole multi-level token $T$ as input. Each feature map extracted by each block of $D$ is added to the features after AdaIN. Combining AdaIN and token decoder D, TPS effectively captures both global style information and detailed high-frequency features, leading to more refined face generation results with rich facial details.

## C MORE IMPLEMENTATION DETAILS

### C.1 TRAINING DETAILS.

Our training process is divided into two stages. In the first stage, we only optimize the geometry encoders, without incorporating the tokenizer and TFS, for coarse FLAME parameters estimation. In this stage, the total loss is set as $\mathcal{L} = \mathcal{L}_{lmk}$. In the second stage, similar to SMIRK Retsinas et al. (2024), we alternately train the encoders (including the geometry encoder and MFAT) and TFS during the training process. This design helps reduce the effect of the neural renderer compensating for the encoders. Specifically, for each iteration, we first freeze the TFS and update the encoders. Then, we freeze the encoders and update the TFS only. By alternating the training of the encoders and TFS, we avoid the joint optimization of these two components. Additionally, the frozen part acts as a regularizer for the other training component, enhancing generalization.

### C.2 EVALUATION METRICS DETAILS

To validate our motivation and demonstrate that the 2D images we reconstructed have higher quality, we utilized several image-level evaluation metrics. Additionally, we also assessed the stability of the reconstructed videos, which is greatly beneficial for subsequent practical application such as face reenactment and identity swapping. **(i) Image Reconstruction:** For image reconstruction quality, the Peak Signal-to-Noise Ratio (**PSNR**) measures the low-level similarity of the generated images to the ground-truth images. The Fréchet Inception Distance (**FID**) Heusel et al. (2017) metric is employed to measure the dissimilarity between distributions of generated and real images. Of particular importance, we compute the cosine similarity of identity (**CSIM**) features to assess the fidelity of identity preservation. These features are derived from the pretrained face recognition model ArcFace Deng et al. (2019a). Furthermore, following the previous work PIRenderer Ren et al., the Average Expression Distance (**AED**) and Average Pose Distance (**APD**) metrics are employed to scrutinize the impact of pose and expression imitation. **(ii) Video Reconstruction:** We use warp-error Geyer et al. (2023) to evaluate the temporally consistency of reconstructed video. We determine the flicker level of the entire video by calculating the average brightness difference between adjacent frames.

### C.3 LOSS DETAILS

Previous methods for single-image face geometry reconstruction often heavily rely on 2D facial landmark detection due to the lack of necessary 3D annotations. These 2D landmarks are used to constrain the accuracy of the projected mesh. However, we have observed that the commonly used 2D facial landmarks are inaccurate for many scenarios. For example, in Fig. A-2, asymmetric mouth shapes or exaggerated facial expressions often lead to erroneous mouth landmarks with widely used detectors such as the 68-point facial detector Feng et al. (2021) or Mediapipe Zielonka et al. (2022a); Retsinas et al. (2024), which in turn degrades the accuracy of face reconstruction. In our experiment, we set $\epsilon = 0.05$ in Eq. 12.

Our investigation revealed that the facial landmark detector provided by InsightFace Guo & Deng (2022) delivers accurate and robust facial keypoints. However, since these keypoints are 2D, they can only detect visible facial contours. The corresponding mesh landmarks are often fixed at specific semantic locations, such as the jawline and the sides of the nose. In cases of large head poses, this leads to significant discrepancies between the 3D mesh landmarks and the detected 2D landmarks. To address this issue, we designed a head-pose-based landmark loss that only constrains the visible mesh landmarks. This provides more precise supervision signals. The landmarks selected under different head poses are dynamically adjusted based on visibility. Therefore, our selected facial keypoints focus on the eyes, nose, and mouth regions. We empirically observed that: 1) Yaw angle significantly affects the accuracy of nose and jawline keypoints, 2) Pitch angle has minimal impact on keypoint loss. Please find more details at the right in Fig. A-2

Although our experiments demonstrate that this loss term brings significant improvements to the model, we used a relatively hard mask to directly determine effective landmarks based on different head poses. Designing a smoothly varying confidence score to apply to all landmarks could potentially make better use of all landmark information, including eyebrows. Therefore, we plan to further explore a stronger landmark loss in the future.

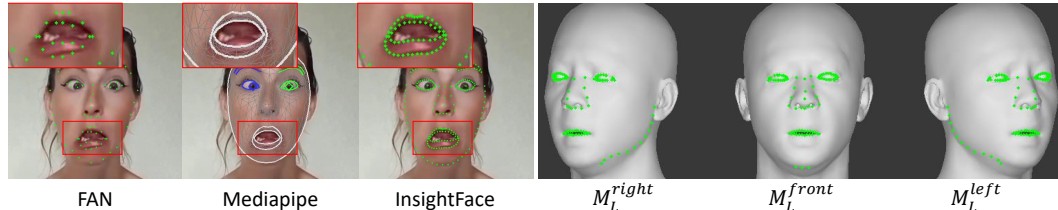

FAN     Mediapipe     InsightFace     $M_L^{right}$     $M_L^{front}$     $M_L^{left}$

Figure A-2: (Left) We use InsightFace Guo & Deng (2022) to detect 203 2D facial landmarks, which get more accurate mouth landmarks. (Right) Different masks for landmark loss under different head poses.

Table A-1: Numerical comparison on 3D facial geometry benchmark.

| Method | NoW benchmark | | | Stirling benchmark | | | FaceWareHouse $\times e2$ | | |
|---|---|---|---|---|---|---|---|---|---|
| | Median ↓ | Mean ↓ | Std. ↓ | Median ↓ | Mean ↓ | Std. ↓ | Median ↓ | Mean ↓ | Std. ↓ |
| Deep3DFace*( Deng et al. (2019b)) | 1.11 | 1.41 | 1.21 | 0.99 | 1.27 | 1.15 | 2.98 | 4.11 | 3.93 |
| 3DDFA-V2 (Guo et al. (2020)) | 1.09 | 1.38 | 1.18 | 1.20 | 1.55 | 1.45 | 3.12 | 3.94 | 3.87 |
| DECA (Feng et al. (2021)) | 1.23 | 1.57 | 1.39 | 1.03 | 1.32 | 1.18 | 3.09 | 4.05 | 3.91 |
| SMIRK (Retsinas et al. (2024)) | 0.99 | 1.22 | 1.02 | 1.01 | 1.08 | 1.05 | 2.87 | 3.98 | 3.89 |
| TEASER (Ours) | 0.92 | 1.10 | 0.99 | 1.00 | 1.07 | 1.04 | 2.78 | 3.87 | 3.81 |

# D    MORE RESULTS

**More results on 3D benchmarks.** In Fig. A-3, we present a comparison of our method with SMIRK on the Stirling benchmark Feng et al. (2018b) for facial expression reconstruction. It is visually evident that our method reconstructs facial meshes with greater accuracy, particularly in terms of mouth closure. Although we did not specifically focus on optimizing the 3D face shape estimation, the improved accuracy of expression reconstruction in our approach allows us to achieve performance on par with other methods on the Stirling benchmark (as reported at Tab. A-1). Both the Stirling and NoW benchmarks primarily focus on face shape reconstruction accuracy, which is not our main contribution, our method still shows competitive performance in Tab.A-1. To better demonstrate our effectiveness on 3D expression reconstruction, we compared our method on FaceWareHouse Cao et al. (2013), a dataset that provides 3D geometry meshes with different expressions across 150 subjects. We used the now evaluation codebase [1] with standard (non-metrical) evaluation to calculate 'median/mean/std' metrics. All images from FaceWareHouse are used for benchmarking. Note that methods marked with * in Table A-1 used additional 3D datasets during training, making direct comparisons with our method unfair. Nevertheless, our method still achieves comparable results.

In Table A-2, we have provided more quantitative comparisons on LRS3 test set with other advanced method under same condition. Our method consistently achieves better performance across metrics.

**More ablation study results.** In Fig. A-4, we demonstrate the effectiveness of the token consistency loss. With this loss, we can reconstruct more faithful facial images, capturing clearer facial details, such as gaze direction, sharpness of the teeth, and finer details of glasses.

**Number of network parameters analysis.** To further analysis our token enhanced face renderer can generated superior face quality, not dependent on more trainable parameters, we experimented with increasing SMIRK's renderer parameters to match or exceed our renderer size. We found that increasing the learnable parameters of the renderer did not significantly improve the performance

---

[1] https://github.com/soubhiksanyal/now_evaluation

Table A-2: More numerical comparisons against SOTA methods.

| Methods | LIPIS ↓ | FID ↓ | CSIM ↑ | PSNR ↑ | AED ↓ | APD ↓ |
|---|---|---|---|---|---|---|
| DECA( Feng et al. (2021)) | 0.442 | 151.6 | 0.209 | 11.90 | 0.151 | 0.057 |
| 3DDFA-v3 (Wang et al. (2024)) | 0.181 | 25.18 | 0.604 | 25.78 | 0.141 | 0.054 |
| SMIRK (Retsinas et al. (2024)) | 0.109 | 25.39 | 0.729 | 29.14 | 0.147 | 0.056 |
| TEASER (Ours) | 0.077 | 19.41 | 0.804 | 30.67 | 0.114 | 0.044 |

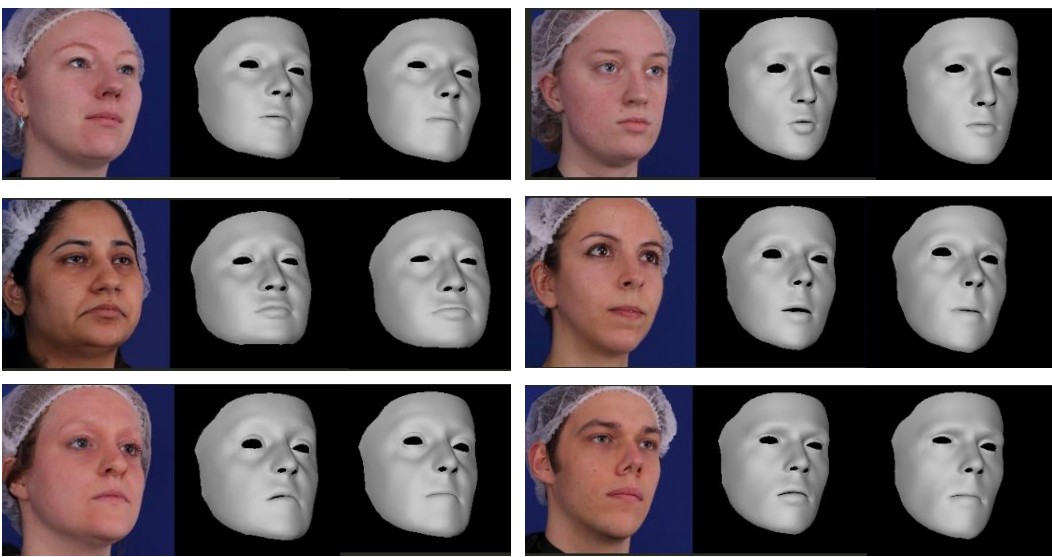

Figure A-3: Visual mesh results on stiring result dataset. From left to right: input images, reconstructed mesh from SMIRK, and ours.

Table A-3: Analysis on number of trainable parameters in our renderer. Our method shows consistently superior performance compared to SMIRK with different scale of model size.

| Methods | # Params. | LIPIS ↓ | FID ↓ | PSNR ↑ | CSIM ↑ | AED ↓ | APD ↓ |
|---|---|---|---|---|---|---|---|
| SMIRK | 31.4M | 0.109 | 25.39 | 0.729 | 29.14 | 0.147 | 0.056 |
| SMIRK (res_block=6) | 36.1M | 0.108 | 25.31 | 0.731 | 29.12 | 0.146 | 0.056 |
| SMIRK (res_block=6, init_feat=40) | 56.4M | 0.108 | 25.32 | 0.728 | 29.16 | 0.147 | 0.055 |
| TEASER (Ours) | 32.7M | 0.077 | 19.41 | 0.804 | 30.67 | 0.114 | 0.044 |

of SMIRK. Even with these larger models, our method still achieves better results, as shown in the Table. A-3.

**Inference speed.** On an RTX 3090, our method runs at an overall speed of 20.43 FPS. Both the Encoder and Face Renderer operate above real-time, with the Encoder at 43.94 FPS and the Token-guided Face Synthesizer at 45.77 FPS. Our model can be easily converted to a ONNX model, after which the overall speed increases to 29.77 FPS.

## E    UNDERSTANDING APPEARANCE TOKENS

In Fig. A-6, we present more interpretable results of the tokens. Using videos corresponding to different identities, we extracted tokens and observed that the extracted tokens cluster effectively around the same identity. This demonstrates that the tokens extracted by TEASER exhibit strong interpretability and disentanglement with respect to identity.

Fig. A-5 illustrates how tokens at different scales affect face appearance. We extract multi-scale appearance tokens from the target image, while the source image provides face geometry. Specifically, $z_1$ represents low-level tokens, and $z_4$ represents high-level tokens. It can be observed that low-level tokens primarily influence skin tone, while high-level tokens affect semantic details such as gender, eyebrow shape, *etc.*

## F    DISCUSSIONS

### F.1    LIMITATIONS AND FUTURE WORK.

The 3D geometry encoder of our method inherits several limitations that are specific to FLAME-based head reconstruction methods. While we do address some of the limitations that stem from

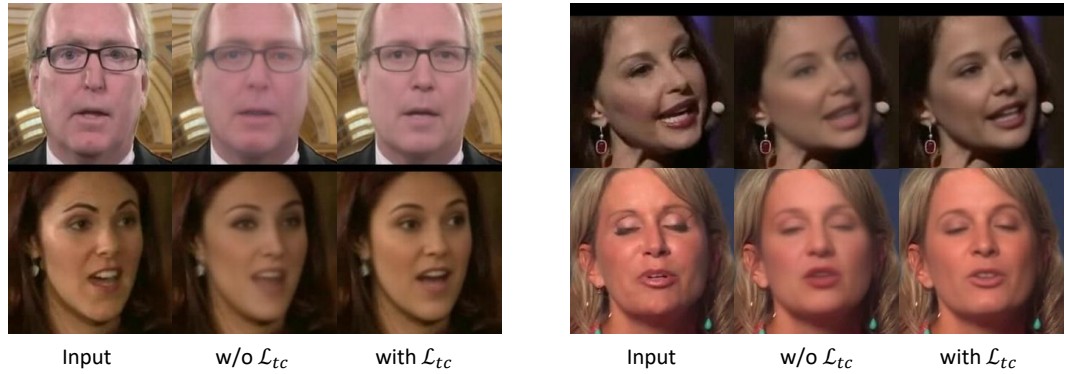

Figure A-4: More results about effectiveness of token consistency loss.

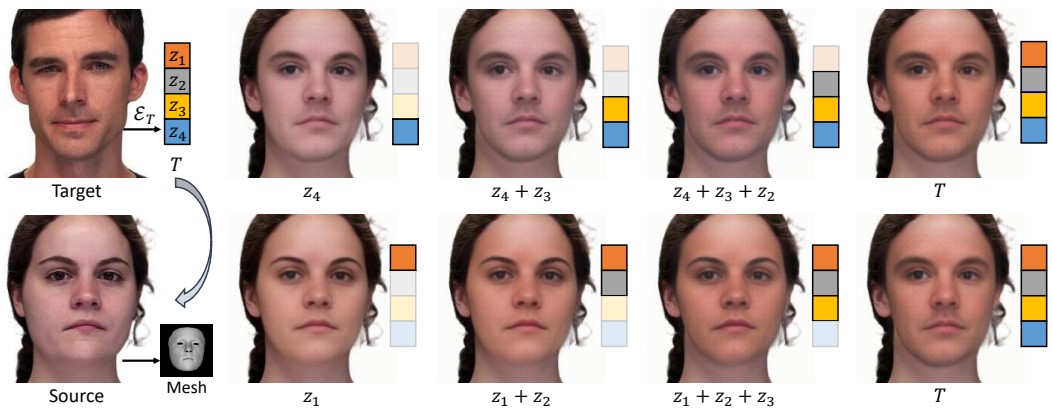

Figure A-5: Understanding how tokens at different scales affect face appearance.

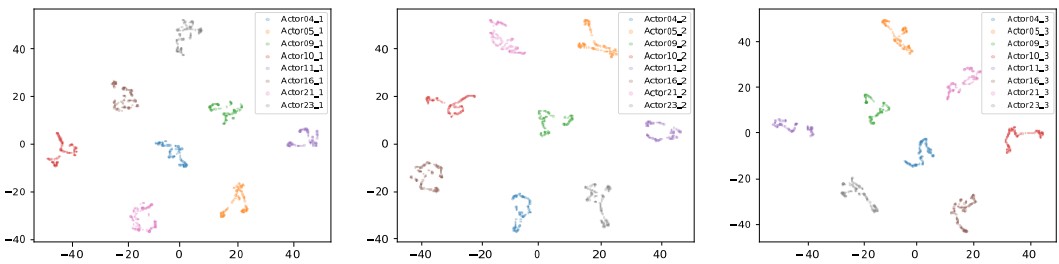

Figure A-6: More results about token visualization.

the use of a parametric FLAME model, we are still subject to misalignment on extreme poses and strongly occluded faces. Our face synthesis stage cannot generate overly complex facial occlusions, such as hands, microphones, bangs, and other facial accessories. Future work can explore using semantic segmentation to filter out non-facial regions when calculating the loss, in order to improve the accuracy and robustness of the synthesis-by-neural-rendering scheme. Our work can also be applied to future SOTA talking head generation tasks, providing accurate expression information and a general facial renderer.

## F.2    ETHICAL CONSIDERATIONS.

This research is conducted with a focus on advancing the accuracy and realism of 3D expression capture technologies for legitimate and ethical applications, such as enhance user experience in VR/AR/XR and improving visual effects in the social media. Specifically, the ability to edit faces and transfer expressions could facilitate the creation of deepfake videos, which might be used to manipulate the likeness of individuals without their consent. This includes generating highly realistic but fabricated content, such as making individuals appear to say or do things they never did or placing their likenesses in misleading or harmful contexts. Such misuse poses risks to privacy, consent, and trust in digital media, as well as broader societal implications, such as spreading misinformation or damaging reputations. We strongly object to the misuse of our method for any harmful or deceptive purposes. Our research is intended to support the scientific community and ethical industry practices. We encourage ongoing dialogue and regulation to ensure that developments in this area are employed in ways that uphold individual rights and align with community values. Future work should also prioritize an in-depth exploration of these issues, including robust mitigation strategies, transparent usage guidelines, and technical safeguards to prevent misuse.

