# OpenReview forum: "TEASER: Token Enhanced Spatial Modeling for Expressions Reconstruction"
_ICLR.cc/2025/Conference — ICLR 2025 Poster_

### Official Review · Reviewer_KnVN · 2024-10-31

**Soundness:** 2
**Presentation:** 3
**Contribution:** 2
**Rating:** 6
**Confidence:** 4

**Summary:**

This work proposes a compact representation of self-occlusion shadows, glasses, lighting and various skin tone into a token. A facial neural renderer that output photo-realistic human portrait conditioning on the tokens. A token cycle loss for self-supervised training of the tokenizer and the neural renderer, and a pose-dependent loss and a region loss for better details reconstruction. Experiments show better image reconstruction quality compared to SMIRK.

**Strengths:**

1. The temporal consistency of the rendered portrait clips is significantly better than SMIRK.
2. The reconstructed facial geometry shows better expression details.

**Weaknesses:**

1. (L063-L069) states the limitations of previous methods: texture color is coupled with illumination, the rendered outputs are over-smoothed, and failed to render highly reflective facial surfaces. However, the results of the proposed method still suffer from these issues. The rendered portrait clips still suffer from over-smoothed textures, lack of highlights and fail to faithfully model the high specular region.
2. Lack of an analysis of the concept represented by the multi-scale appearance tokenizer. i.e. which scale corresponds to the semantics and which scale corresponds to the texture details? What is the definition of the ‘semantics’? What would happen if we only change certain scales? Compared to previous methods which use simple render pipelines, the interpretability of the current method is hindered by the reliance on the deep neural renderer. Thus a complete analysis of the MSAT is crucial to determine to what extend can the pipeline be interpreted.
3. Though targeted at 3D facial reconstruction, this work completely drops metrics related to the face geometry, relying solely on the 2D image metrics. The argument (L393-L395) is possibly correct, no experiments are conducted to validate it. Running some experiments on a dataset with ground-truth 3D geometry can possibly support it.

**Questions:**

1. What contributes to the better temporal consistency?
2. How to measure the matching between the geometry and the rendered face? A reason previous method use simple renderer is because this mapping is defined explicitly. However here the mapping is implicit and there is no explicit textures obtained for downstream tasks.
3. (L372-L374) Training: the pre trained geometry encoder contains the shape, expression and head-pose encoder. Do the shape and head-pose encoder keep frozen during the entire training process, and only the expression encoder is updated? If this is the case, can we say that the contributions to better shape-matched facial rendering all come from the neural facial renderer, rather than the geometry shape estimator?

---

> ### Author Response · Authors · 2024-11-24
> **Author Response to Reviewer KnVN**
>
> Thank you for your valuable comments and acknowledgement of our effectiveness. We hope our response fully resolve your concerns.
>
> > Q1: The rendered portrait clips still suffer from over-smoothed textures.
>
> To clarify, our motivation is to provide more accurate photometric supervision for 3D expression reconstruction.
> Reconstructing high-frequency details for each image using a general model is inherently challenging. As shown in Fig. 4, our method can better reconstruct and capture highlights and shadows compared to other methods. However, rendering high specular regions and heavily occluded areas while maintaining image sharpness remains challenging, as discussed in our limitations Sec.. We will investigate this further in future work. Thank you.
>
> > Q2: Lack of an analysis of the concept represented by the multi-scale appearance tokenizer.
>
> Thank you for your meaningful suggestion. We have demonstrated in Table 2 that using multi-scale tokens improves our method's performance.
> To further analyze how different scales of appearance tokenizer affect face reconstruction, we designed an experiment where we sequentially swap appearance tokens of different scales between two individuals. Please refer to **Fig A-5** in the revised paper for the details.
> We found that 1）Shallow-level appearance tokens relate to facial texture details (e.g., skin tone) and 2) Deep-level appearance tokens capture high-level semantics (e.g., gender, shape of eyebrow, etc.)
> This observation aligns with findings from previous work (VGG loss, ResNet).
> Additionally, our renderer generates facial motions conditioned on the input face mesh. Since multi-scale appearance tokens don't contain facial geometry information, our reconstructed face images maintain consistency with the input mesh geometry. We demonstrate this capability for various facial expressions in our [supplementary video](https://openreview.net/attachment?id=pTeOOKnjGM&name=supplementary_material)(at 1min 7s).
>
> > Q3: Running some experiments on a dataset with ground-truth 3D geometry can possibly support it.
>
> Thank you very much for pointing out these. In the Appendix Table A-1, we compare our method with additional approaches on the Stirling benchmark for 3D geometry reconstruction, where we achieve comparable or better results. We have also validated our method on the NoW benchmark and FaceWareHouse dataset, demonstrating superior performance on both. More details please refer to Reviewer `N9UC` Q1.
>
> > Q4: What contributes to the better temporal consistency?
>
> Our method achieves better temporal consistency through two main factors:
> 1. Better reconstruction of facial details
> 2. Stable appearance token estimation
>
> Additionally, we use only one token (first frame) for expression-driven face video generation.
>
> > Q5: How to measure the matching between the geometry and the rendered face?
>
> We evaluate the matching in following ways:
> 1. We employ various 2D metrics to measure shape and expression consistency between reconstructed and input faces.
> 2. We validate using datasets with ground truth 3D facial scans.
>
> > Q6: Can we say that the contributions to better shape-matched facial rendering all come from the neural facial renderer, rather than the geometry shape estimator?
>
> No, it cannot. Our renderer primarily conditions on the facial mesh and appearance tokens, which do not include geometric information about the face (for more details, please refer to Reviewer `N9UC` Q3).
>
> The facial shape and expressions in the renderer come entirely from the facial mesh, which is obtained by rendering the FLAME mesh produced by the encoder. Its accuracy of facial geometry directly impacts the results of the renderer. In addition, we adopt a token-cycle loss and an iterative training strategy to reduce the effect of the neural renderer compensating for the encoders. Furthermore, our experimental results on several 3D benchmarks also validate that our method improves geometry shape.
>
> **Summary**
>
> Based on your suggestions, we have added more analysis of the different level appearance tokens to Fig. A-5 and corresponding analysis to Sec. E. We think this enhances the paper and better explains the method. Thank you again for your valuable comments.

---

> > ### Comment · Reviewer_KnVN · 2024-11-26
> >
> > Thanks for the comments. The additional experiments on 3D geometry reconstruction are important and promising. I suggest the authors emphasize them in the main paper instead of the appendix. I decided to raise my score.
> > However, I prefer a marginal acceptance because the results still suffer from over-smoothed textures.
> > Thanks again for the hard work and responsiveness!

---

> > > ### Author Response · Authors · 2024-11-28
> > >
> > > We are grateful for your acknowledgment of our efforts, which is truly encouraging. We will update the main paper as suggested. Thank you again for raising the score.

---

### Official Review · Reviewer_N9UC · 2024-11-03

**Soundness:** 2
**Presentation:** 3
**Contribution:** 2
**Rating:** 6
**Confidence:** 3

**Summary:**

This paper proposes a method for accurate facial reconstruction from single images. The central contribution of this method is a token-based neural renderer, that the authors argue helps improve geometry reconstruction by minimizing rendering errors. The neural renderer takes as an image, from which features are extracted using a CNN encoder. Different layers of this CNN are treated as a multi-scale feature bank which are projected to a common dimensional space by an MLP and concatenated together to generate the final appearance feature.  The appearance feature is then used to condition a UNet that renders the final image. In order to ensure the appearance tokens do not leak any information a token consistency loss is used. Additional losses to ensure rendering fidelity are specified in equation 10 of the paper. Both quantitative and qualitative results show improvements over prior work. However, I believe that in its current form, the evaluation is incomplete.

**Strengths:**

1) The paper is easy to follow and well written
2) The intuition of having a more powerful renderer for better reconstruction is, I believe, the right one.
3) Both qualitative and quantitative results show improvements over prior art.

**Weaknesses:**

1) Right now the paper only reports image reconstruction error and contains no measure of ground-truth reconstruction accuracy. Since the renderer is conditioned on the input image, it is possible that the geometric reconstruction itself is not that great but the renderer just learns to copy the appearance pixels from the tokens. There is no evaluation in the paper to demonstrate that this does not happen. Ideally, I would’ve likes to see at least some ground-truth measure of error, for example, using the NoW dataset. In this case, two images of different views can be chosen at random, one can be used to condition the neural renderer and the other can be treated at the RGB ground-truth. Then both the geometric reconstruction error and the RGB error can be measured independently. This would make sure the renderer is actually learning the appearance tokens and not just copying the input pixels.

2) In general the sensitivity of the appearance tokens to different expressions/views of the same person must be evaluated. There is hardly any analysis of that.

3) While using multi-scale features does make sense, I would imagine the SMIRK renderer with more parameters would be able to learn them without explicit inductive bias. A general parameter count comparison with the SMIRK renderer would help place the current contributions in context.

**Questions:**

I have summarized by criticism and questions in points 1) and 2) of weaknesses. In general I would like to see greater analysis of the robustness of the renderer to difference expressions and views of the same person.

---

> ### Author Response · Authors · 2024-11-24
> **Author Response to Reviewer  N9UC (1/2)**
>
> Thank you for your valuable comments and acknowledgement of our novelty and effectiveness. We hope our response fully resolve your concerns.
>
> > Q1: Contains no measure of ground-truth reconstruction accuracy.
>
> Thank you for your reminder. First, we have provided comparison results on the Stirling benchmark in our Appendix, where our method achieves comparable results with SOTA methods.
> Second, following the reviewer's suggestion, we evaluated our method on the NoW benchmark, where it also performs well (results shown in the table below).
> Note that both the Stirling and NoW benchmarks primarily focus on face shape reconstruction accuracy, which is not our main contribution, our method still shows competitive performance.
> Finally, we compared our method on [FaceWareHouse](http://kunzhou.net/zjugaps/facewarehouse/), a dataset that provides 3D geometry meshes with different expressions across 150 subjects. We used the [now_evaluation](https://github.com/soubhiksanyal/now_evaluation) codebase with standard (non-metrical) evaluation to calculate `median/mean/std` metrics. All images from FaceWareHouse are used for benchmarking. Methods marked with * in below table used additional 3D datasets during training, making direct comparisons with our method unfair. Nevertheless, our method still achieves comparable results.
>
> | Methods |	NoW benchmark $\downarrow$ |	Stirling benchmark $\downarrow$ |	FaceWareHouse ($\times e2$) $\downarrow$ |
> | - | - | - | - |
> | Deep3DFace* |	1.11/1.41/1.21 |	0.99/1.27/1.15 |	2.98/4.11/3.93 |
> | DECA |	1.09/1.38/1.18 |	1.03/1.32/1.18 |	3.12/3.94/3.87 |
> | 3DDFA-v2 |	1.23/1.57/1.39 |	1.20/1.55/1.45 |	3.09/4.05/3.91 |
> | SMIRK |	0.99/1.22/1.02 |	1.01/1.08/1.05 |	2.87/3.98/3.89 |
> | Ours |	**0.92**/**1.10**/**0.99** |	1.00/**1.07**/**1.04**	| **2.78**/**3.87**/**3.81** |
>
> > Q2: This would make sure the renderer is actually learning the appearance tokens and not just copying the input pixels.
>
> A2: We have designed our method specifically to prevent the renderer from simply copying input pixels, specifically:
>
> 1. Our token features are spatially flattened, which eliminates spatial information from the input images.
> 2. We design token cycle loss to enforce the learned token robust to different facial expressions.
> 3. We alternately train the encoders (including the geometry encoder and MFAT) and TFS during the training process. This design helps reduce the effect of the neural renderer compensating for the encoders.
> 4. In our [supplementary video](https://openreview.net/attachment?id=pTeOOKnjGM&name=supplementary_material) (at 40s), we have demonstrated that our method can generate stable facial video with one fixed token, which are extracted from the first frame.
>
> For more training details and loss details, please refer to Sec. B.1.
>
> > Q3: In general the sensitivity of the appearance tokens to different expressions/views.
>
> In fact, we considered this situation. To reduce the sensitivity of the appearance token to different expressions, we designed the following approach and conducted validation.
> 1. We introduced a cycle token loss with expression augmentation to ensure tokens remain consistent for the same person across different expressions.
> 2. To enhance appearance token stability and prevent tokens from simply copying input pixels, we spatially flatten the token features, which eliminates spatial information from input images.
> 3. Our experiments validate the disentanglement capabilities, showing that tokens can be used for downstream tasks, including generating facial videos with new expressions and views.
> 4. For better understanding, we visualized the features in Fig. 8, demonstrating that our learned appearance tokens cluster by identity and are minimally affected by different expressions/views.

---

> ### Author Response · Authors · 2024-11-24
> **Author Response to Reviewer N9UC (2/2)**
>
> > Q4: Imagine the SMIRK renderer with more parameters would be able to learn them without explicit inductive bias.
>
> Thanks for your excellent comment. Our renderer has only a modest increase in parameters compared to SMIRK's renderer. We experimented with increasing SMIRK's renderer parameters to match or exceed our model size. We found that increasing the learnable parameters of the renderer did not significantly improve the performance of SMIRK. Overall, our method achieves significantly better results, as shown in the table below:
>
> | Methods |	params |	LIPIS $\downarrow$ |	FID $\downarrow$ |	CSIM $\uparrow$ |	PSNR $\uparrow$ |	AED $\downarrow$	| APD $\downarrow$ |
> | - | - | - | - | - | - | - | - |
> | SMIRK |	31.4M |	0.109 |	25.39 |	0.729 |	29.14 |	0.147 |	0.056 |
> | SMIRK (res_block=6) |	36.1M |	0.108 |	25.31 |	0.731 |	29.12 |	0.146 |	0.056 |
> | SMIRK (res_block=6, init_feat=40) |	56.4M |	0.108 |	25.32 |	0.728 |	29.16 |	0.147 |	0.055 |
> | Ours (TEASER)	 | 32.7M |	**0.077** |	**19.41** |	**0.804** |	**30.67** |	**0.114** |	**0.044** |
>
> **Summary**
>
> Thank you for your valuable comments. Following your advices, we have added the quantitative comparisons on three 3D face reconstruction benchmarks and discussions to Sec. D in Appendix. We have also added number of network parameters analysis to Sec. D in appendix. We believe such discussions and updates makes the paper more complete and better explains the method. Thank you again for your valuable feedback.

---

> > ### Comment · Reviewer_N9UC · 2024-11-26
> >
> > I would like the thank the authors for their response. They have answered my question satisfactorily and looking at the other reviews and responses I have decided to raise my score.

---

> > > ### Author Response · Authors · 2024-11-28
> > >
> > > Thank you very much for recognizing our work. Following your insightful suggestions, we have added 1) the quantitative comparisons on three 3D face reconstruction benchmarks and 2) number of network parameters analysis to our paper. We believe such discussions and updates makes the paper more complete and clearer.

---

### Official Review · Reviewer_VmdW · 2024-11-04

**Soundness:** 3
**Presentation:** 4
**Contribution:** 3
**Rating:** 8
**Confidence:** 4

**Summary:**

The authors in this paper propose a hybrid face representation for better 3D face reconstruction and associated applications. The facial geometry is represented by FLAME and appearance is represented by a multiscale appearance token, which are combined using a token-guided neural renderer to generate reconstructed mesh with high fidelity. The paper also introduces novel loss functions and achieves SOTA results, which are adequately substantiated by ablation studies and experimental analysis.

**Strengths:**

1) The incorporation of the multiscale appearance token in the generator is novel. I like the idea of combining AdaIN and Controlnet based feature incorporation and the reasoning provided by the authors for the same.
2) The details provided in the supplementary are really helpful to better understand the method (and makes the method implementable by the research community). The video results also prove some of the claims in the paper (like temporal stability).

**Weaknesses:**

The paper is heavily based on the architecture / methodology of SMIRK, so some novel contributions of this paper are marginal and/or inspired by other existing methods.

**Questions:**

1) The authors mentioned using MobilenetV3 for performance and efficiency. Since efficiency is considered, what is the inference execution time of the proposed approach?
2) How is the parameter $\epsilon$ tuned in equation 13?

---

> ### Author Response · Authors · 2024-11-24
> **Author Response to Reviewer VmdW**
>
> We are grateful for your positive review and valuable comments, and we hope our response fully address your concerns.
>
> > Q1: The paper is heavily based on the architecture / methodology of SMIRK
>
> A1: We acknowledge SMIRK's valuable insight that *analysis-by-neural-synthesis supervision* is essential for accurate 3D expression reconstruction. However, our method differs significantly in several technical aspects:
>
> 1. We designed a hybrid representation for 3D expression reconstruction and introduced a more accurate facial renderer.
> 2. We introduce our method with several novel components, including 1) multi-scale tokens, 2) token cycle loss, 3) pose-dependent keypoint loss.
> 3. Our approach achieves both 1) Accurate 3D expression mesh extraction and 2) Implicit and interpretable facial appearance representation.
> 4. Compared to SMIRK, our method not only supports multiple downstream tasks but also generates more stable and high-quality portrait reconstructions. Furthermore, our method enables better video driving results
>
> Therefore, we believe these differences clearly distinguish our work from SMIRK including architecture and methodology.
>
> > Q2: What is the inference execution time of the proposed approach?
>
> A2: On an RTX 3090, our method runs at an overall speed of 20.43 FPS. Both the Encoder and Face Renderer operate above real-time, with the Encoder at 43.94 FPS and the Token-guided Face Synthesizer at 45.77 FPS.
> Our model can be easily converted to a ONNX model, after which the overall speed increases to 29.77 FPS.
>
> > Q3: How is the parameter $\epsilon$ tuned in equation 13?
>
> A3: We compared the keypoints obtained from the FLAME mesh with the pseudo keypoints estimated by InsightFace to determine at what head pose there is a significant difference. We empirically set the parameter $\epsilon = 0.05$ by observing face images with various image with different headposes.
>
> **Summary**
>
> Regarding your comments, we add detailed inference time to Section D, and add $\epsilon$ settings to the main paper. Please find the updated contents in the revision. We highlight new contents with **[blue color]()**. We think such updates enhance the paper and better explain the method. Thank you again for your positive rating and valuable feedback.

---

> > ### Comment · Reviewer_VmdW · 2024-11-30
> >
> > I would like to thank the authors for addressing my concerns and updating the submission accordingly. After reading other reviews and overall rebuttal, I have decided to stick to my initial rating.

---

> > > ### Author Response · Authors · 2024-12-02
> > >
> > > We are delighted that our response has successfully addressed your concerns. We appreciate your recognition of our work and the constructive comments you provided, which have made our paper more complete.

---

### Official Review · Reviewer_x2jZ · 2024-11-05

**Soundness:** 3
**Presentation:** 3
**Contribution:** 3
**Rating:** 8
**Confidence:** 4

**Summary:**

This paper proposes a new model for reconstructing facial expressions more accurately from a single image by combining explicit facial parameters with an implicit, multi-scale appearance token. The method includes a token-guided neural renderer that captures detailed facial features, such as shadows, lighting, and subtle expressions, and a token cycle loss for self-supervised training, enhancing stability and interpretability for downstream applications such as face editing and expression transfer.

**Strengths:**

+ The paper is well-written and the main aspects of the proposed methodology are clearly presented.

+ The introduction of the appearance token (main novelty) seems to provide some advantages over the closely-related method of SMIRK (Retsinas et al. 2024). These are mostly related to the photorealism of the synthesized face images

+ The experimental evaluation is done on two datasets (LRS3 and HDTF) and shows that the proposed method achieves improvements in terms of estimation of the detailed 3D geometry of the face.

+ The paper includes also a detailed ablation study comparing 6 different versions of the proposed pipeline and providing evidence that all components are indeed useful.

**Weaknesses:**

- The novelty over the closely-related method of SMIRK (Retsinas et al. 2024) is rather limited. The only crucial addition is the appearance tokenizer. A careful inspection of the results reveals that this addition seems to have fairly small impact on the quality of the 3D face reconstructions.

- In terms of methodology, the pose-dependent masking of eq. (13) seems to be suboptimal. It is an overly hard masking, since it creates discontinuities when the yaw angle becomes +/- \epsilon (a smoothly-varying confidence score for every landmark would have been more appropriate). In addition, the masking doesn't take into account the pitch angle at all, which is unrealistic since variation of the pitch angle results to occluded landmarks too.


- From the description of Section 4, it seems that the same individuals of the LRS3 are used in both training of the method and test (Table 1). Even if different frames of the individuals are used in training and testing, this is an unfair and inadequate experimental protocol.

- The paper includes quantitative as well as qualitative comparisons with only two recent SOTA methods of 3D face reconstruction (SMIRK and 3DDFA-V3), which is insufficient.  Other methods such as DECA, EMOCA, Deep3DFace and FOCUS could have been included.

- The paper presents results of applying the proposed method to the downstream tasks of Identity Swapping and Face Animation (Figs. 6 and 7). However, there is no comparison with SOTA methods, which is inadequate.

**Questions:**

- Please comment on the novelties and advantages over the closely-related method of SMIRK.

- Please comment on the pose-dependent masking.

- Please clarify the issue regarding the LRS3 dataset (training and testing) that I mentioned above.

- Please comment on my criticism regarding the insufficient experimental comparisons (in terms of 3D face reconstruction as well as the downstream tasks of identity swapping and face animation).

**Details Of Ethics Concerns:**

The paper presents how the proposed method for 3D reconstruction of facial expressions can be used for face editing and expression transfer. This can be used for the generation of deepfake videos without the consent of the depicted individuals, making them say things that they've never said and placing their faces in video scenes where they've never been, all in a highly realistic manner. There are several ethic concerns related to these application, which need to be taken seriously into account. Section D.2 in the Appendix includes a relevant discussion, but it is not sufficiently detailed, provided the associated ethical risks.

---

> ### Author Response · Authors · 2024-11-24
> **Author Response to Reviewer x2jZ (1/2)**
>
> We are grateful for your positive review and valuable comments. We hope our response fully resolves your concerns.
>
> > Q1: About the novelty over the closely-related method of SMIRK.
>
> We first apologize for any lack of clarity in our previous explanation. We acknowledge that SMIRK indeed inspired our work. However, there are many significant differences in several technical aspects:
>
> 1. We designed a hybrid representation for 3D expression reconstruction and introduced a more accurate facial renderer.
> 2. We introduce our method with several novel components:
>     a) Multi-scale appearance tokens
>     b) Expression-independent token cycle loss
>     c) Pose-dependent keypoint loss
> 3. Our approach achieves both a) Accurate 3D expression mesh extraction and  b) Implicit and interpretable facial appearance representation
> 4. Compared to SMIRK, our method:
>     a) Supports multiple downstream tasks
>     b) Generates more stable and high-quality portrait reconstructions
>     c) Enables better [video driving](https://openreview.net/attachment?id=pTeOOKnjGM&name=supplementary_material) results
>
> We believe these differences clearly distinguish our work from SMIRK. In addition, our experimental results demonstrate better performance, including:
> - Significant improvements in 3D face expression reconstruction
> - Enhanced stability in [video portrait reconstruction](https://openreview.net/attachment?id=pTeOOKnjGM&name=supplementary_material) (7-38s)
> - Better performance on the Stirling benchmark. We have also demonstrated more competitive performance on NoW benchmark and FaceWareHouse benchmark (Please refer to Reviewer `N9UC`-Q1). We appreciate that other reviewers have also acknowledged these improvements.
>
>
> > Q2: About the pose-dependent masking of eq. (13).
>
> Thanks your for the suggestion. Our initial focus was on demonstrating that this loss term brings significant improvements to the model. Our experiments show that it enhances both 1) overall model performance, and 2) local facial details, particularly around the mouth corners and eyes. Since our selected facial keypoints focus on the eyes, nose, and mouth regions, we observed that:
> - Yaw angle significantly affects the accuracy of nose and jawline keypoints, please see Fig. A-2 for more details.
> - Pitch angle has minimal impact on keypoint loss.
>
> Therefore, we think such design is effective. We agree that using a smoothly-varying confidence score is a reasonable suggestion. We will investigate this further in future work. Thank you.
>
>
> > Q3: About experimental protocol.
>
> We first apologize for the misunderstanding. We clarify that we did not use videos of the same individuals for both training and testing. Following previous methods, our training and testing samples come from different videos with no overlap between them. So we believe the comparison is fair and adequate. We will make this clearer in the revised paper.
>
> > Q4: The paper includes quantitative as well as qualitative comparisons with only two recent SOTA methods of 3D face reconstruction. Other methods could have been included
>
> Thanks for your advise. We have provided visual comparisons with several SOTA methods in Fig. 3-4.
> Following your suggestion, we have included more quantitative comparisons with other methods, shown below. Our method consistently achieves better performance across metrics:
>
> | Methods | LPIPS $\downarrow$ | FID $\downarrow$ | CSIM $\uparrow$ | PSNR $\uparrow$ | AED $\downarrow$ | APD $\downarrow$ |
> | - | - | - | - | - | - | - |
> | DECA	         | 0.422 |	151.6 |	0.209 | 11.90 |	0.151 |	0.057 |
> | 3DDFA-v3	     | 0.181 |	56.18 |	0.604 | 25.78 |	0.141 |	0.054 |
> | SMIRK	         | 0.109 |	25.39 |	0.729 | 29.14 |	0.147 | 0.056 |
> | TEASER(Ours)	 | **0.077** |	**19.41** |	**0.804**| **30.67** |	**0.114** |	**0.044** |
>
> We have also present comparisons with additional methods on three 3D facial benchmark in Appendix Table A-1, where our method achieves overall better performance in 3D face shape reconstruction.
>
> > Q4: About comparison with SOTA methods for downstream tasks.
>
> Thanks for your suggestion. The main purpose of our downstream tasks is to demonstrate the disentanglement capabilities of our learned hybrid representation. Specifically:
> - Face mesh can explicitly drive facial movements
> - Appearance tokens can accurately swap facial appearances
>
> More comparisons with SOTA methods specifically designed for these downstream tasks will be our future work.

---

> ### Author Response · Authors · 2024-11-24
> **Author Response to Reviewer x2jZ (2/2)**
>
> > Q5: More discussion about ethics risks.
>
> Thanks for your constructive suggestions. The proposed method for 3D reconstruction of facial expressions, while innovative, raises significant ethical concerns due to its potential misuse. Specifically, the ability to edit faces and transfer expressions could facilitate the creation of deepfake videos, which might be used to manipulate the likeness of individuals without their consent. This includes generating highly realistic but fabricated content, such as making individuals appear to say or do things they never did or placing their likenesses in misleading or harmful contexts. Such misuse poses risks to privacy, consent, and trust in digital media, as well as broader societal implications, such as spreading misinformation or damaging reputations. Although Section D.2 in the Appendix addresses these concerns, the discussion is insufficiently detailed given the gravity of the ethical risks. Future work should prioritize an in-depth exploration of these issues, including robust mitigation strategies, transparent usage guidelines, and technical safeguards to prevent misuse.
>
> **Summary**
>
> Following your comments, we add more design details and discussions to the pose-guided landmark loss. Due to page limitations, we have included more quantitative comparisons with DECA, 3DDFA-v2 to Table A-2 in Appendix. Finally, we add more discussions about ethics risks to the ethical considerations. Please find more details in the revised paper, in which the updated contents are highlighted with **[blue color]()**. Thank you again for your suggestions and positive feedback for our paper.

---

> > ### Comment · Reviewer_x2jZ · 2024-11-26
> >
> > I'd like to thank the authors for their responses. These helped in the clarification of several issues and addressed my relevant concerns. Also, there was some expansion of the experimental evaluation by including quantitative comparisons with DECA, which show that the proposed method has clear advantages over DECA too.
> > Importantly, reading the other reviewers' comments and the rebuttal, I am now more positive about the novelty of the proposed method.
> > For the above reasons, I am raising my rating for this paper.

---

> > > ### Author Response · Authors · 2024-11-28
> > >
> > > Thank you very much for acknowledging our work; your recognition is truly uplifting. We also appreciate your insightful suggestions, which have further enhanced the quality of our work and offered us valuable perspectives.

---

### Author Response · Authors · 2024-11-24
**Common Response**

We thank all reviewers for their valuable feedback and for acknowledging that our paper is well written (`x2jZ/N9UC`), clearly presented (`x2jZ`), provides advantages (`x2jZ`), mostly related to photorealism (`x2jZ`), achieves improvements (`x2jZ/VmdW/N9UC/KnVN`), and that the multiscale appearance token is novel (`VmdW`), like the idea, the right one(`VmdW/N9UC`), supplementary are really helpful(`VmdW/KnVN`).

We have carefully considered the comments and will take them into account to further enhance the quality of our work. We have updated the corresponding PDF and listed the updates from the paper here.

- We include experimental results for evaluating our method on two 3D facial geometry benchmarks in **Table A-1** in the Appendix.
- We add more design details and discussions about the pose-guided landmark loss to the Appendix (Sec. C.3).
- We add more quantitative comparisons with DECA and 3DDFA-v2 to the Appendix (please find more details in **Table A-2**).
- We include an analysis of the number of network parameters in Sec. D and **Table A-3** in the Appendix.
- We conduct experiments to understand how tokens at different scales affect face appearance in **Fig. A-5**.
- In the Appendix, we add more discussions about ethical risks in the ethical considerations section.

For better reading experience, we have also enclosed a PDF highlighting the updates in the [supplementary files](https://openreview.net/attachment?id=pTeOOKnjGM&name=supplementary_material). In this PDF, the updated content is distinguished using **[blue text]()**. We believe these updates make the paper more complete.

We also remain committed to addressing any further questions or concerns from the reviewers promptly.

Best regards,
The Authors

---

### Meta-Review · Area_Chair_ufBe · 2024-12-22

**Metareview:**

The paper tackles the problem of facial expression reconstruction. Initially, while giving overall positive scores, the reviewers listed a number of questions to the authors. They included questions about novelty, methodology, further comparisons, and other questions and concerns. The authors addressed these concerns to a large extent, by providing additional comparisons (with DECA), explanations about the contribution of the work and clarification of the results. While the paper received quite solid average score, the AC believes that the review of VmdW, doesn't contain sufficient support to verify the high rating given, instead, it looked a bit superficial. The decision is therefore to accept the manuscript. Congrats!

**Additional Comments On Reviewer Discussion:**

The authors did a good job addressing the comments of the reviewers. After the discussion period the scores either went up or stayed the same.

---

### Decision · Program_Chairs · 2025-01-22

Accept (Poster)